# Hemodialysis and Plasma Oxylipin Biotransformation in Peripheral Tissue

**DOI:** 10.3390/metabo12010034

**Published:** 2022-01-04

**Authors:** Tong Liu, Inci Dogan, Michael Rothe, Julius V. Kunz, Felix Knauf, Maik Gollasch, Friedrich C. Luft, Benjamin Gollasch

**Affiliations:** 1Experimental and Clinical Research Center (ECRC), Charité Medical Faculty and Max Delbrück Center (MDC) for Molecular Medicine, 13125 Berlin, Germany; tong.liu@charite.de (T.L.); maik.gollasch@med.uni-greifswald.de (M.G.); friedrich.luft@charite.de (F.C.L.); 2LIPIDOMIX GmbH, Robert-Rössle-Str. 10, 13125 Berlin, Germany; inci.dogan@lipidomix.de (I.D.); michael.rothe@lipidomix.de (M.R.); 3Department of Nephrology and Medical Intensive Care, Charité—Universitätsmedizin Berlin, Augustenburger Platz 1, 13353 Berlin, Germany; julius-valentin.kunz@charite.de (J.V.K.); felix.knauf@charite.de (F.K.); 4Department of Internal Medicine and Geriatrics, University Medicine Greifswald, 17475 Greifswald, Germany; 5HELIOS Klinikum Berlin-Buch, Schwanebecker Chaussee 50, 13125 Berlin, Germany

**Keywords:** hemodialysis, eicosanoids, lipidomics, oxylipins, erythrocyte, arterio-venous, biotransformation

## Abstract

Factors causing the increased cardiovascular morbidity and mortality in hemodialysis (HD) patients are largely unknown. Oxylipins are a superclass of lipid mediators with potent bioactivities produced from oxygenation of polyunsaturated fatty acids. We previously assessed the impact of HD on oxylipins in arterial blood plasma and found that HD increases several oxylipins. To study the phenomenon further, we now evaluated the differences in arterial and venous blood oxylipins from patients undergoing HD. We collected arterial and venous blood samples in upper extremities from 12 end-stage renal disease (ESRD) patients before and after HD and measured oxylipins in plasma by LC-MS/MS tandem mass spectrometry. Comparison between cytochrome P450 (CYP), lipoxygenase (LOX), and LOX/CYP ω/(ω-1)-hydroxylase metabolites levels from arterial and venous blood showed no arteriovenous differences before HD but revealed arteriovenous differences in several CYP metabolites immediately after HD. These changes were explained by metabolites in the venous blood stream of the upper limb. Decreased soluble epoxide hydrolase (sEH) activity contributed to the release and accumulation of the CYP metabolites. However, HD did not affect arteriovenous differences of the majority of LOX and LOX/CYP ω/(ω-1)-hydroxylase metabolites. The HD treatment itself causes changes in CYP epoxy metabolites that could have deleterious effects in the circulation.

## 1. Introduction

Survival rates among end-stage renal disease (ESRD) hemodialysis (HD) patients are poor, and excess death rate is related to cardiovascular disease [1,2]. Lipids are essential for many functions in the body, where they serve as integral components for cellular membranes as well as energy storage and signaling molecules [3]. Polyunsaturated fatty acids (PUFA) are metabolized by different enzymes, mainly cytochromes P450 (CYP), monooxygenase, cyclooxygenase (COX), and lipoxygenase (LOX)/CYP ω/(ω-1)-hydroxylase pathways, which can produce a large superclass of biologically active substances, namely oxylipins (Figure 1).

Arachidonic acid (AA)-derived oxylipins, especially those derived from the CYP450 pathway, produce vasodilation and exhibit pro-angiogenic, anti-inflammatory, and cardioprotective effects [5]. Docosahexaenoic acid (DHA)-derived 19, 20-epoxydocosapentaenoic acid (EDP) and eicosapentaenoic acid (EPA)-derived 17, 18-epoxyeicosatetraenoic acid (EEQ) exert anti-arrhythmic effects by inhibiting Ca^+^ and isoproterenol-induced increase in cardiomyocyte contractility [6]. Beyond, many biological functions of this large group of widely still unknown lipids are undiscovered. Moreover, the main metabolic pathway of oxylipins is that a major part is hydrolyzed by soluble epoxide hydrolase (sEH) to biologically less-active diols [7]. The other part is re-esterified into phospholipids as a temporary storage pool of oxylipins that can be rapidly mobilized to exert biological effects when the organism receives a stimulus [8]. Another important source of circulating esterified oxylipins are lipoprotein-bound oxylipins, for example, very low-density lipoproteins (VLDL), which bind to cell surface lipoprotein lipases when they reach tissue cells [9]. In any case, esterification is important both in also removing free oxylipins signals and in enabling direct effects of esterified oxylipins. Currently, their specific mechanisms of action are largely unknown.

In previous studies, we established the lipidomics approach for the analysis of oxylipins in human blood. We demonstrated that there are specific patterns of oxylipins profiles in peripheral blood released during short-term maximal cardiovascular stress (stress ergometry) that may influence cardiovascular function [10,11]. We also observed that hemodialysis treatment increased plasma levels of CYP epoxy-metabolites but did not change the majority of LOX/CYP ω/(ω-1)-hydroxylase metabolites [4]. The changes primarily affected esterified metabolites, whereas no significant differences were observed in free metabolites [4].

The arteriovenous oxygen difference is the difference in the oxygen content of the blood between the arterial blood and the venous blood. Knowing this difference aids in clinical diagnostics of how much oxygen is removed from the blood in capillaries as the blood circulates in the body. Uremic patients undergoing hemodialysis present pre-HD with high ammonia levels in arterial blood with a significantly positive arteriovenous difference [12]. The arteriovenous blood sugar content is due to the fact that the peripheral tissues, especially the muscles, either store or burn part of the glucose that passes through them [13]. The same is true for non-esterified fatty acids [14], which can develop positive or negative arteriovenous differences depending on patient’s physical or health status and organs perfused [13,14]. Moreover, arteriovenous differences in plasma NO_2_^−^ levels (as an index of endothelial nitric oxide (NO) formation) can detected in NO_2_^−^ loading conditions and exercise [15].

To gain information on lipid biotransformation, we collected arterial and venous blood samples in upper extremities from ESRD patients and tested the hypothesis that hemodialysis affects the arteriovenous difference of these metabolites (Figure 2). We performed the experiments to better understand biotransformation of the metabolites in vivo, particularly whether or not the peripheral tissues, especially the muscles in the upper limbs, either produce, store, or degrade part of the epoxy-metabolites that pass through them. We were particularly interested to clarify which arteriovenous relationships are present in plasma oxylipin profiles during hemodialysis and in what way they are modified by extracorporeal renal replacement therapy, which causes oxidative stress, chronic inflammation, and red blood cell–endothelial interactions (cf. circled arrows in Figure 2).

## 2. Results

### 2.1. Clinical Characteristics

The clinical features of ESRD hemodialysis (HD) patients are summarized in Table 1. The patients had focal segmental glomerulosclerosis (six patients), ADPKD (autosomal dominant polycystic kidney disease) (one patient), IgA nephropathy (one patient), hypertensive nephropathy (one patient), renal amyloidosis (one patient), drug-induced kidney injury (one patient), and cystic kidneys (one patient). All patients had major cardiovascular complications, such as cardiovascular and cerebrovascular events, and/or peripheral artery disease. Appendix A shows that the patients were not diabetic but had hyperlipidemia.

### 2.2. Effects of Hemodialysis

#### 2.2.1. Pre-HD

The effects of hemodialysis treatment on plasma oxylipins in our patients are shown (Table 2 and Appendix A). With exception of 11-HETE and 13-HODE, comparison between total CYP, LOX and LOX/CYP ω/(ω-1)-hydroxylase metabolites levels from arterial and venous blood showed no arteriovenous differences before HD treatment (pre-HD), i.e., the levels of individual CYP epoxy-metabolites were not significantly different in arterial vs. venous blood (Table 2). These metabolites included 5,6-EET, 8,9-EET, 11,12-EET, 14,-15-EET, 5,6-DHET, 8,9-DHET, 11,12-DHET, 14,15-DHET, 7,8-EDP, 10,11-EDP, 13,14-EDP, 16,17-EDP, 19,20-EDP, 7,8-DiHDPA, 10,11-DiHDPA, 13,14-DiHDPA, 16,17-DiHDPA, 19,20-DiHDPA, 5,6-EEQ, 8,9-EEQ, 11,12-EEQ, 14,15-EEQ, 17,18-EEQ, 5,6-DiHETE, 8,9-DiHETE, 11,12-DiHETE, 14,15-DiHETE, 17,18-DiHETE, 9,10-EpOME, 12,13-EpOME, 9,10-DiHOME, 12,13-DiHOME, 5-HETE, 8-HETE, 9-HETE, 12-HETE, 15-HETE, 4-HDHA, 7-HDHA, 8-HDHA, 10-HDHA, 11-HDHA, 13-HDHA, 14-HDHA, 16-HDHA, 17-HDHA, 20-HDHA, 5-HEPE, 8-HEPE, 9-HEPE, 11-HEPE, 12-HEPE, 15-HEPE, 18-HEPE, 9-HODE, 16-HETE, 17-HETE, 18-HETE, 19-HETE, 20-HETE, 22-HDHA, and 20-HEPE (Table 2).

Figure 1 shows that the main pathway of EET, EpOME, EEQ, and EDP biotransformation in many cells is conversion to DHETs, DiHOMEs, DiHETEs, and DiHDPAs. This is achieved by the soluble epoxide hydrolase enzyme (sEH). Since ESRD might have caused EET, EpOME, EEQ, and EDP production rapidly degraded to their diols, we next analyzed the sums of the individual CYP epoxy-metabolites and their diols (Table 3). We found that ESRD was not associated with differences in the sums in arterial vs. venous blood; i.e., there was no arteriovenous differences in the total levels of the metabolites (Table 3). Moreover, we calculated diol/epoxide ratios of the epoxy-metabolites (Table 3) and found that the four classes of epoxy-metabolites are unequally hydrolyzed to appear in the arterial circulation. We found that EEQs are better metabolized into their diols (ratio of DiHETEs/EEQs; 0.6541 ± 0.4424) than EETs, EDPs, and EPOMEs (ratios of those diols/epoxy-metabolites, 0.1104 ± 0.0878, 0.1369 ± 0.1095, and 0.2017 ± 0.0845, respectively; Dunn’s multiple comparison test, *p* > 0.05) (Table 3). In fact, the following order of ratios was identified: DiHETEs/EEQs > DiHOMEs/EpOMEs = DiHDPA/EDPs = DHETs/EETs (Dunn’s multiple comparison test, *p* < 0.05). This pattern was also found for the individual metabolites in the venous blood, as shown (Table 3). Together, the findings indicate that CYP epoxy-metabolites are unequally hydrolyzed by sEH in arterial and venous blood in vivo [4]. However, there is no arteriovenous difference in the epoxy-metabolites before hemodialysis.

#### 2.2.2. Post-HD

Similar to our previous study [4], hemodialysis treatment caused an increase in a number of epoxy-metabolites in arterial blood, including 5,6-EET, 8,9-EET, 11,12-EET,14,15-EET, 8,9-EEQ, 11,12-EEQ, 9,10-EpOME, 12,13-EpOME, 9-HODE, and 13-HODE. The levels of other CYP, LOX, and LOX/CYP ω/(ω−1)-hydroxylase metabolites were unchanged (Appendix A).

However, the effects were more obvious in venous blood. Here, we first compared the individual CYP epoxy-metabolites and their diols (Appendix A). We observed that hemodialysis increased 5,6-EET, 8,9-EET, 11,12-EET, 14,15-EET, 5,6-DHET, 8,9-DHET, 7,8-EDP, 10,11-EDP, 13,14-EDP, 16,17-EDP, 19,20-EDP, 5,6-EEQ, 8,9-EEQ, 11,12-EEQ, 14,15-EEQ, 17,18-EEQ, 5,6-DiHETE, 9,10-EPOME, 12,13-EPOME, 9,10-DiHOME, 12,13-DiHOME, 5-HETE, 12-HETE, 12-HEPE, 9-HODE, 13-HODE, and 22-HDHA levels in venous blood (Appendix A). The levels of other CYP, LOX, and LOX/CYP ω/(ω−1)-hydroxylase metabolites were unchanged (Appendix A). As next step, we calculated the sums of the individual CYP epoxy-metabolites and their diols (Table 3) and compared the levels between arterial and venous blood (Table 3). The data show that the four classes of CYP metabolites (i.e., EET, EpOME, EEQ, and EDP plus their respective diols) were more prominently accumulated in venous blood compared to arterial blood (Table 3); i.e., we detected negative arteriovenous differences in 5,6-EET, 8,9-EET, 11,12-EET, 14,15-EET, 8,9-DHET, 11,12-DHET, 7,8-EDP, 10,11-EDP, 14,15-EDP, 16,17-EDP, 19,20-EDP, 7,8-DiHDPA, 10,11-DiHDPA, 5,6-EEQ, 8,9-EEQ, 11,12-EEQ, 14,15-EEQ, 17,18-EEQ, 9,10-EPOME, 12,13-EPOME, 12-HETE, and 17-HDHA (Table 2). This decrease was not seen for free epoxy-metabolites, with the exception of 5,6-DiHETE and 14,15-EET (Table 4). Thus, it is unlikely that the decrease in arteriovenous differences of total epoxy-metabolites is caused by metabolites in free state. The arteriovenous differences of other CYP, LOX, and LOX/CYP ω/(ω−1)-hydroxylase metabolites were unchanged (Table 2).

Together, the findings indicate that hemodialysis increases all four classes of CYP eicosanoids, particularly in venous blood, to cause negative arteriovenous differences of a number of CYP metabolites after the dialysis treatment (post-HD).

### 2.3. Diol/Epoxide Ratios and sEH Activity

To clarify possible mechanisms underlying this preferred increase in venous blood, we calculated diol/epoxide ratios of the eicosanoids (Appendix A). Our analysis of sums of the individual CYP epoxy-metabolites and their diols demonstrated increased accumulation of all four classes of CYP epoxy-metabolites (5,6-EET + 5,6-DHET, 8,9-EET + 8,9-DHET, 11,12-EET + 11,12-DHET, 14,15-EET + 14,15-DHET, 7,8-EDP + 7,8-DiHDPA, 10,11-EDP + 10,11-DiHDPA, 13,14-EDP + 13,14-DiHDPA, 16,17-EDP + 16,17-DiHDPA, 19,20-EDP + 19,20-DiHDPA, 5,6-EEQ + 5,6-DiHETE, 8,9-EEQ + 8,9-DiHETE, 11,12-EEQ + 11,12-DiHETE, 14,15-EEQ + 14,15-DiHETE, 17,18-EEQ + 17,18-DiHETE, 9,10-EpOME + 9,10-DiHOME, 12,13-EpOME + 12,13-DiHOME, 5,6-DHET/5,6-EET, 8,9-DHET/8,9-EET, 11,12-DHET/11,12-EET, and 14,15-DHET/14,15-EET) in venous blood (Appendix A). We found that ratios of diols/epoxides of all four subclasses were reduced by dialysis (Appendix A). The effects were more obvious in venous compared to arterial blood (Table 3). Together, the results indicate that decreased sEH activity in peripheral tissue, especially the muscles, in vivo contributes to release and accumulation of CYP epoxy-metabolites in the peripheral upper limb circulation during hemodialysis.

## 3. Discussion

To gain further insight into oxylipin metabolism, we evaluated the arteriovenous differences of oxylipins levels in uremic patients treated by HD treatment. Our major findings are three-fold, as follows: (1) We detected negative arteriovenous differences of a number of CYP metabolites (EETs, EDPs, EEQs, EPOMEs, and their diols) after the dialysis treatment (post-HD). These were primarily due to exuberant increases of the metabolites in the venous blood stream of the arm. However, hemodialysis did not change arteriovenous differences of the majority of LOX and LOX/CYP ω/(ω-1)- hydroxylase metabolites. (2) The observed arteriovenous differences were caused by the dialysis treatment itself since we did not detect arteriovenous differences between the CYP metabolite levels before HD treatment (pre-HD). (3) Decreased soluble epoxide hydrolase (sEH) activity contributed to the release and accumulation of the CYP metabolites after HD. Together, our data indicate that CYP epoxy-metabolites are influenced by renal-replacement therapies and are consistent with the notion that blood perfusing peripheral tissue, especially the muscle, acts as a stimulus for release and accumulation of CYP epoxy-metabolites in the upper limb during dialysis treatment. Based on our data, we suggest that CYP epoxy-metabolites may be a contributing factor to the blood flow response in the peripheral circulation during hemodialysis. Future studies can clarify whether the identified metabolites exhibit beneficial or detrimental cardiovascular effects, possibly in metabolite-interacting networks.

Fick first capitalized on the arteriovenous oxygen difference to determine cardiac output and thereby to determine oxygen delivery and extraction. His insights permitted measuring the amount of oxygen taken up from the blood by the individual tissues. Usually, the arterial oxygen concentration is measured in blood from the radial, femoral, or brachial artery, and the oxygen content of mixed venous blood is measured from blood collected from the right heart (i.e., pulmonary artery). The mixed venous oxygen content represents the weighted average of oxygen content in venous blood from all organ systems. However, venous blood can also be withdrawn from specific organs, e.g., legs, arms, or the mesentery. In this case, measurement of oxygen consumption by the specific organ system requires withdrawal of venous blood draining that organ [12,13,14,15]. We applied this approach to better understand biotransformation of oxylipins in vivo, particularly whether or not the peripheral tissues, especially the muscles in the upper limbs, either produce, store, or degrade part of the epoxy-metabolites that pass through them in response to dialysis treatment. We were particularly interested to understand how the plasma oxylipins are modified by extracorporeal renal replacement therapy, which is known to cause oxidative stress, chronic inflammation, and red blood cell–endothelial interactions. Our study identified negative arteriovenous differences of a number of CYP metabolites in upper extremities, especially the muscles, after the dialysis treatment (post-HD). These changes were caused by exuberant increases of all four subclasses of the CYP epoxy-metabolites (i.e., EET, EDP, EEQ, EpOME) in the venous blood stream of the upper limbs.

We found that the following CYP epoxy-metabolites were increased in venous blood after HD: 5,6-EET, 8,9-EET, 11,12-EET, 14,15-EET, 7,8-EDP, 10,11-EDP, 14,15-EDP, 16,17-EDP, 19,20-EDP, 5,6-EEQ, 8,9-EEQ, 11,12-EEQ, 14,15-EEQ, 17,18-EEQ, 9,10-EPOME, and 12,13-EPOME. Although reduced in-vivo sEH activity in CKD/ESRD [4,16] may have contributed to the increased accumulation of the four eicosanoid classes itself, our data indicate that the observed changes are likely related to reduced sEH activity by HD treatment. Accordingly, we found that ratios of diols/epoxides of all four subclasses were reduced by dialysis. The sEH is an enzyme that converts specific epoxides to less biologically effective diols [7]. In plasma, we were able to observe that all ratios of diols/epoxides were reduced in the venous plasma after hemodialysis. Going further, we compared the differences in arterial and venous levels before and after dialysis and found that almost all the differences in metabolite ratios were due to their decrease in venous plasma after dialysis. Nevertheless, whether or not dialysis treatment inhibits the hydrolysis of epoxide metabolites to diols by sEH remains to be experimentally verified.

Although CYP450 can oxidize AA to four regional isomers, their content is unevenly distributed, with 11,12- and 14,15-EET being the predominant regional isomers in mammals, accounting for approximately 67–80% of all EETs [17,18]. Endothelial and red blood cells store EETs and can release these epoxides into plasma [19,20,21,22]. The mechanisms of how EETs and other epoxy-metabolites are released from the tissues are largely unknown, making our findings difficult to explain. The anti-inflammatory effects of EETs in the cardiovascular system, in the kidney, and in the brain are well established, and their main vascular effect is the diminution of the NF-kappaB-dependent inflammatory response [23,24]. 5,6-EET can produce vasodilation in various vascular beds [25,26]. 8,9-EET inhibits NF-kappaB nuclear translocation in mouse B lymphocytes activated by lipopolysaccharide, causing a reduction in basal and activation-induced antibody production [27]. As a result of suppressing LOX-1 receptor upregulation and NF-kappaB activation, 11,12 and 14,15-EET reduce the oxidized low density lipoprotein (LDL)-related inflammation in the vasculature [28]. 5,6-EET is the only one of these regional isomers that induces a bidirectional effect. In a study on the pulmonary vasculature [29], 5,6-EET was found to cause sustained vasoconstriction in the lung during hypoxia. Our findings are consistent with the idea that EETs are relevant vasodilatory signaling molecules for exhibiting cardiovascular effects in ESRD/CKD [30], which could counteract vasoconstrictor responses during dialysis. Pharmaceutical sEH inhibition is viewed as a novel treatment for enhancing the beneficial biological effects of EETs and other epoxy-metabolites [5]. However, presumably higher levels of EETs in blood and tissue in vivo may have also detrimental cardiovascular effects [31,32,33]. The extent to which the EETs increase observed in our study has detrimental or beneficial biological effects should be investigated.

We detected increases in 7,8-EDP, 10,11-EDP, 14,15-EDP, 16,17-EDP, 19,20-EDP, 5,6-EEQ, 8,9-EEQ, 11,12-EEQ, 14,15-EEQ, and 17,18-EEQ during dialysis. Little is known about the (patho) physiological functions of EEQs and EDPs. 16,17-EDP and 19,20-EDP are potent vasodilators in the peripheral circulation. They decrease blood pressure and can protect the heart by preservation of mitochondrial function [19,34]. EEQs and EDPs have a significant role in anti-cardiac fibrosis, improving post-ischemic reperfusion injury in the heart and reducing inflammation and pain, especially 19,20-EDPs [35] and 17,18-EEQs [36]. EDPs protect cardiac cells by improving and maintaining mitochondrial function against LPS-induced cell damage, as shown in a recent study [37]. A recent mouse ex vivo experiment [35] found that 19, 20-EDP could exert cardioprotective effects by inhibiting NLR Family Pyrin Domain Containing 3 (NLRP3) inflammatory vesicles compared to 17,18-EEQ. 17,18-EEQ decreases endothelial activity and prevents atherosclerosis [19]. 17,18-EEQ has been identified to activate Ca^2+^-activated potassium channels to produce vasodilation, an effect that is even more effective than for EETs [38,39]. 17,18-EEQ also inhibits Ca^+^ and isoproterenol-induced increases in cardiomyocytes contractility, suggesting that 17,18-EEQ could be used as a potential antiarrhythmic agent [6]. Our findings implicate that both EEQs and EDPs are novel candidates for vasoactive substances potentially released by hemodialysis to affect hemodynamics in these conditions.

We observed pronounced increases in venous plasma levels of 9,10-EpOME and 12,13-EpOME after hemodialysis. Recent findings suggest that EpOMEs exhibit cardio-inhibitory effects [40,41,42] and vasodilation [43]. Furthermore, they can cause vasoconstrictor effects in ischemic heart disease [40,44]. Since we observed increases in 9,10-EpOME and 12,13-EpOME after hemodialysis, our findings suggest the notion that increases in EpOMEs could play a detrimental role in cardiac ischemia and hemodynamics in ESRD patients.

We believe that our findings could have clinical relevance. We were able to analyze arteriovenous relationships in plasma oxylipin profiles in ESRD patients and the extent to which these are altered by renal replacement therapies. Our data indicate that CYP epoxy-metabolites are influenced by the hemodialysis treatment and are consistent with the notion that dialyzed blood perfusing peripheral tissue, especially the muscle, acts as a stimulus for release and accumulation of CYP epoxy-metabolites. Our analysis allowed an overall assessment of the biotransformation of plasma oxylipins in peripheral tissue, specifically upper extremity muscle, in vivo. The results provided new insights into local metabolic and hemodynamic formation processes of oxylipins in peripheral tissues in vivo. The extent to which the observed effects in oxylipins cause detrimental or beneficial cardiovascular effects, possibly in metabolite-interacting networks, or are influenced by specific underlying renal diseases or patient phenotypes should be explored in future studies.

## 4. Materials and Methods

### 4.1. Participants

The ethical committee of the Charité University Medicine approved the study. Written informed consent was obtained from all participants. The study was duly registered on the ClinicalTrials.gov website (Identifier: NCT03857984). Nine men and three women were recruited. To be included in the study, age over 18 years and presence of CKD requiring hemodialysis (three times a week) with stable hemodialysis prescriptions were defined as inclusion criteria. They had to have been dialyzed through a native fistula or a gore-tex graft. Exclusion criteria included age less than 18 years, pregnancy, inability to follow simple instructions, non-compliance with their dialysis prescription, and an anemia with hemoglobin (Hb) below 8.0 g/dL or an active infection.

### 4.2. Assessment

All patients were treated in sitting position. Subjects underwent dialysis treatments with a Polyflux 170H dialyzer (PAES membrane, Gambro); the ultrafiltration rate was unchanged during the hemodialysis treatment. Relevant dialysis treatment conditions (blood flow rate with ~250 mL/min, dialysate flow rate ~500 mL/min, double needle puncture technique, and dialysis time on average 4 h 15 min) were identical in all dialysis sessions. Arterial (shunt) blood samples were withdrawn on the fistula arm right before beginning of the dialysis (pre-HD) and at the end of the dialysis (5–15 min before cessation, post-HD). Venous blood was collected on the ipsilateral extremity by subcutaneous arm vein puncture at same time points to determine the arteriovenous difference of the epoxy-metabolites (Figure 2). The arteriovenous difference is caused by the fact that the peripheral tissues, specially the muscles, either produce, store, or degrade part of the epoxy-metabolites that perfuse them. All blood samples were obtained by 4 °C precooled EDTA vacuum extraction tube systems. Glucose, cholesterol, and triglycerides were determined in a certified clinical laboratory.

### 4.3. Determination of Eicosanoid Profiles

For the detection of total plasma oxylipins, we took a plasma sample (200 μL), added 300 μL of 10M sodium hydroxide (NaOH), and subjected it to alkaline hydrolysis at 60 °C for 30 min. The sample pH was then adjusted to 6 using 300 μL 58% acetic acid. The prepared samples were then subjected to solid phase extraction (SPE) using a Varian Bond Elut Certify II column. Specific experimental steps were described as previously [11,45]. For the detection of free plasma oxylipins, SPE extraction was performed directly after pH adjustment without prior alkaline hydrolysis.

The extracted metabolites were evaluated by LC-MS/MS using an Agilent 6460 Triple Quad mass spectrometer (Agilent Technologies, Santa Clara, CA, USA) and an Agilent 1200 high-performance liquid chromatography (HPLC) system (degasser, binary pump, well plate sampler, thermostatic column chamber). A Phenomenex Kinetex column (150 mm 2.1 mm, 2.6 m; Phenomenex, Aschaffenburg, Germany) was used in the HPLC system. The specific analysis process has been described previously [45]. Free and total plasma oxylipins were measured in the blood samples.

### 4.4. Statistical Analysis

Descriptive statistics were obtained, and variables were checked for skewness and kurtosis to ensure that they met the normal distribution assumptions. In order to determine statistical significance, paired *t*-test or paired Wilcoxon test were used to compare pre-HD vs. post-HD values. The significance level (*p*) of 0.05 was selected. All data are provided as mean ± SD. The statistics was performed using SPSS Statistics software (IBM Corporation).

## Figures and Tables

**Figure 1 metabolites-12-00034-f001:**
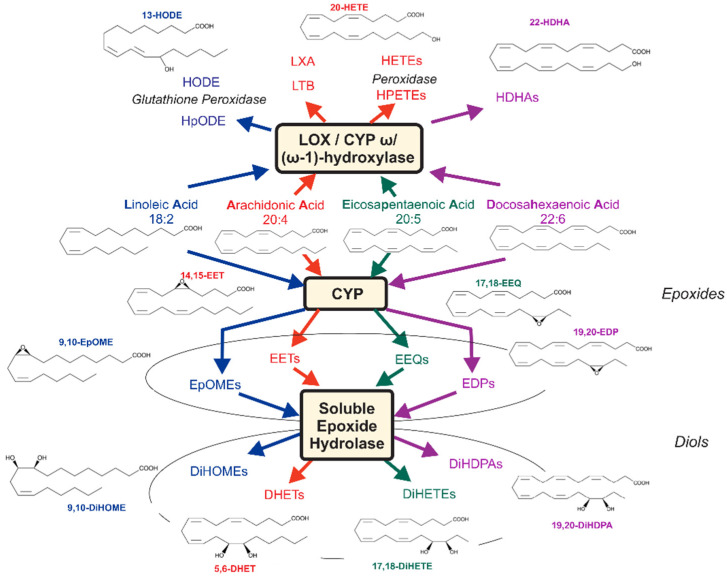
Assessment of cytochrome P450 epoxygenase (CYP) and 12- and 15-lipoxygenase (LOX)/CYP (omega-1)-hydroxylase pathways in response to hemodialysis treatment. Arachidonic (AA), linoleic (LA), eicosapentaenoic (EPA), and docosahexaenoic acids (DHA) are converted to epoxyoctadecenoic acids (EpOMEs, e.g., 9,10-EpOME), epoxyeicosatetraenoic acids (EEQs, e.g., 17,18-EEQ), epoxyeicosatrienoic acid (EETs, e.g., 5,6-EET), and epoxydocosapentaenoic acids (EDPs, e.g., 19,20-EDP) by CYP epoxygenase, respectively. EpOMEs, EETs, EEQs, and EDPs are primarily converted to dihydroxyctadecenoic acids (DiHOMEs), dihydroxyeicosatrienoic acids (DHETs, e.g., 5,6-DHET), dihydroxyeicosatetraenoic acids (DiHETEs, e.g., 17,18-DiHETE), and dihydroxydocosapentaenoic acids (DiHDPAs, e.g., 19,20-DiHDPA) by the soluble epoxide hydrolase (sEH). LA, EPA, AA, and DHA are converted to hydroperoxylinoleic acids (HpODEs), hydroxyoctadecadienoic acids (HODEs), leukotriene B (LTB), lipoxin A (LXA), hydroxydocosahexaenoic acids (HDHAs), hydroperoxyeicosatetraenoic acids (HPETEs), and hydroxyeicosatetraenoic acids (HETEs) by LOX, CYP omega/(omega-1)-hydroxylase, and peroxidase pathways. The metabolites measured in these metabolic pathways follow the changes observed in LA, AA, EPA, and DHA metabolism, respectively. Modified from [4].

**Figure 2 metabolites-12-00034-f002:**
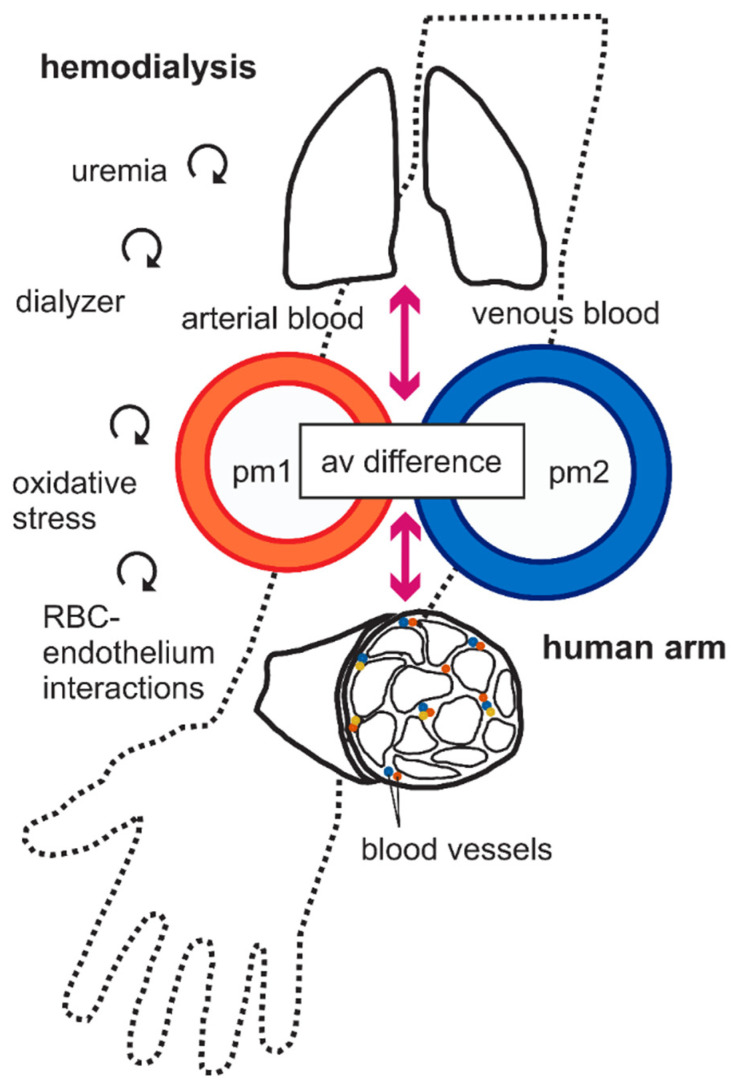
A simplified scheme of the relationship among different compartments. Central and peripheral compartments are shown. Central compartment is consisting of plasma, pm1 and pm2. Peripheral compartments are consisting of organ tissues, especially arm muscle, with extracellular fluid, red blood cells (RBCs), etc. The continuity between arterial and venous systems via pulmonary (top) and peripheral (bottom) arm muscle is illustrated in the diagram. Arterial and venous blood samples were taken before (pre-HD) and after HD (post-HD) treatment. Plasma oxylipins were measured in pm1 and pm2, i.e., arterial shunt and subcutaneous vein, respectively. It is obvious that the arteriovenous (av) difference is caused by the circumstance that the peripheral tissues, especially the muscles, either produce, store, or degrade part of the oxylipins that pass through them. The curved vector graphs represent the hypothetical influence of CKD and hemodialysis in conjunction with shear stress, dialyzer, red blood cell (RBC)-endothelial interactions, and oxidative stress affecting plasma oxylipin biotransformation levels in peripheral tissues.

**Table 1 metabolites-12-00034-t001:** Clinical features of hemodialysis (HD) patients (*n* = 12).

	HD Patients
Age (years)	72 ± 12
Sex	
Male (*n*)	9
Female (*n*)	3
Body mass index (kg/m^2^)	27 ± 3
Race (*n*)	Caucasian = 12
Cause of end-stage renal disease	
Focal segmental glomerulosclerosis (*n*)	6
IgA nephropathy (*n*)	1
Renal amyloidosis (*n*)	1
Hypertension (*n*)	1
Drug induced (*n*)	1
ADPKD (*n*)	1
Cystic kidneys (*n*)	1
Complications	
Cardiovascular (*n*)	12

Notes: Data are presented as mean ± SD or frequencies.

**Table 2 metabolites-12-00034-t002:** Effects of hemodialysis on total oxylipins in the CKD patients before (pre-HD) and at cessation (post-HD) of hemodialysis (*n* = 12 each).

Amount ng/mL	Pre-HD A (Mean ± SD)	Pre-HD V (Mean ± SD)	*p*-Value, *t*-Test (# Paired Wilcoxon Test)	Pre-HD A-V Difference (Mean ± SD)	Post-HD A (Mean ± SD)	Post-HD V (Mean ± SD)	*p*-Value, *t*-Test (# Paired Wilcoxon Test)	Post-HD A-V Difference (Mean ± SD)
**CYP epoxy-metabolites**								
5,6-EET	25.2403 ± 22.8032	16.8013 ± 15.3681	0.325	8.1596 ± 22.0848	**37.0725 ± 16.6846**	**61.8386 ± 21.0905**	**0.003 #**	**−24.7661 ± 17.4189**
8,9-EET	10.5227 ± 10.8994	7.2706 ± 6.6772	0.806	3.1316 ± 11.4525	**13.3085 ± 5.6504**	**22.3969 ± 7.8620**	**0.002 #**	**−9.0884 ± 6.7881**
11,12-EET	9.9539 ± 10.0802	7.0358 ± 6.8420	0.806	2.7042 ± 10.2183	**14.2304 ± 6.6762**	**25.6937 ± 9.1845**	**0.002 #**	**−11.4633 ± 7.4968**
14,15-EET	12.5615 ± 12.5408	9.6655 ± 9.4944	0.538	2.6286 ± 12.4557	**19.0872 ± 9.2999**	**35.6871 ± 13.5842**	**0.002 #**	**−16.6000 ± 10.9006**
5,6-DHET	1.1450 ± 0.5549	1.0727 ± 0.5911	0.389	0.1175 ± 0.1770	1.3684 ± 0.5945	1.2782 ± 0.6163	0.177 #	0.0901 ± 0.2165
8,9-DHET	1.7434 ± 1.5788	1.6063 ± 1.3596	0.806	0.1581 ± 0.4252	**1.6086 ± 1.1218**	**1.2535 ± 0.9769**	**0.004 #**	**0.3550 ± 0.3980**
11,12-DHET	0.4366 ± 0.2207	0.4188 ± 0.1959	0.902	0.0311 ± 0.0878	**0.5221 ± 0.1967**	**0.4683 ± 0.1888**	**0.005**	**0.0538 ± 0.0534**
14,15-DHET	0.4159 ± 0.0996	0.3721 ± 0.0788	0.252	0.0480 ± 0.0902	0.4919 ± 0.1251	0.4212 ± 0.1045	0.132	0.0708 ± 0.1505
7,8-EDP	4.5236 ± 4.3719	3.2930 ± 3.1818	0.951	1.1887 ± 4.7923	**6.0064 ± 2.6897**	**10.4413 ± 3.5636**	**0.002 #**	**−4.4349 ± 2.2351**
10,11-EDP	4.4784 ± 4.4009	3.3370 ± 3.0863	0.806	1.0517 ± 4.5614	**6.6357 ± 3.0803**	**11.9332 ± 3.9274**	**<0.001**	**−5.2975 ± 2.5243**
13,14-EDP	3.3860 ± 3.4990	2.6621 ± 2.8208	0.806	0.6491 ± 3.6773	**4.8226 ± 2.2061**	**7.8177 ± 2.6177**	**<0.001**	**−2.9952 ± 1.9444**
16,17-EDP	2.9692 ± 2.8814	2.1926 ± 2.0367	0.806	0.7541 ± 2.8457	**3.8342 ± 1.6915**	**5.5352 ± 2.0151**	**0.008**	**−1.7010 ± 1.8424**
19,20-EDP	5.5558 ± 5.3859	3.9685 ± 3.1487	1	1.4739 ± 5.6060	**8.6761 ± 3.8647**	**16.3124 ± 6.2669**	**0.002 #**	**−7.6363 ± 3.7823**
7,8-DiHDPA	0.4489 ± 0.2473	0.4110 ± 0.2301	0.498	0.0405 ± 0.0429	**0.4905 ± 0.2287**	**0.4461 ± 0.2152**	**0.021**	**0.0444 ± 0.0571**
10,11-DiHDPA	0.1277 ± 0.0387	0.1210 ± 0.0346	0.664	0.0081 ± 0.0169	**0.1439 ± 0.0452**	**0.1291 ± 0.0539**	**0.031**	**0.0148 ± 0.0207**
13,14-DiHDPA	0.0892 ± 0.0274	0.0846 ± 0.0203	0.65	0.0051 ± 0.0114	0.0989 ± 0.0354	0.1040 ± 0.0439	0.55	−0.0051 ± 0.0285
16,17-DiHDPA	0.0926 ± 0.0361	0.0947 ± 0.0314	0.886	−0.0012 ± 0.0111	0.1160 ± 0.0464	0.1042 ± 0.0480	0.168	0.0118 ± 0.0277
19,20-DiHDPA	0.8379 ± 0.4051	0.8262 ± 0.3658	0.943	0.0383 ± 0.0710	0.9534 ± 0.4118	0.9134 ± 0.4345	0.393	0.0400 ± 0.1559
5,6-EEQ	0.0051 ± 0.0039	0.0034 ± 0.0026	0.23	−0.0064 ± 0.0048	**0.0084 ± 0.0052**	**0.0148 ± 0.0050**	**0.004 #**	**0.0012 ± 0.0042**
8,9-EEQ	1.4478 ± 1.2289	0.9972 ± 0.6972	0.498	0.4347 ± 1.3829	**2.6063 ± 1.7732**	**4.3902 ± 1.5836**	**0.008 #**	**−1.7839 ± 1.6485**
11,12-EEQ	0.9541 ± 0.8310	0.7197 ± 0.4820	0.806	0.2294 ± 0.9263	**1.8077 ± 1.2340**	**3.4951 ± 1.3735**	**0.003 #**	**−1.6874 ± 1.1790**
14,15-EEQ	0.9436 ± 0.7146	0.7047 ± 0.4467	0.343 #	0.2231 ± 0.8184	**1.7544 ± 1.2152**	**3.2728 ± 1.3192**	**0.004 #**	**−1.5184 ± 1.2985**
17,18-EEQ	1.6849 ± 1.3879	1.3915 ± 0.8490	0.543 #	0.2583 ± 1.4915	**2.9621 ± 2.1045**	**6.2605 ± 2.2755**	**0.002 #**	**−3.2984 ± 2.0268**
5,6-DiHETE	1.5299 ± 0.9299	1.3695 ± 0.7134	0.622 #	0.3200 ± 0.8708	2.0357 ± 1.4971	1.6895 ± 0.8335	0.53 #	0.3461 ± 1.2824
8,9-DiHETE	0.0908 ± 0.0378	0.0908 ± 0.0317	0.667 #	0.0061 ± 0.0374	0.1085 ± 0.0552	0.0960 ± 0.0386	0.36	0.0125 ± 0.0455
11,12-DiHETE	0.0476 ± 0.0379	0.0356 ± 0.0100	0.806	0.0128 ± 0.0418	0.0596 ± 0.0600	0.0372 ± 0.0119	0.099 #	0.0224 ± 0.0633
14,15-DiHETE	0.0528 ± 0.0201	0.0462 ± 0.0135	0.364 #	0.0075 ± 0.0205	0.0538 ± 0.0249	0.0487 ± 0.0203	0.49	0.0051 ± 0.0248
17,18-DiHETE	0.2149 ± 0.0825	0.2141 ± 0.0836	0.981 #	0.0030 ± 0.0659	0.2784 ± 0.1176	0.2454 ± 0.1082	0.253	0.0330 ± 0.0948
9,10-EpOME	33.0227 ± 31.0831	22.5987 ± 12.9602	0.806	9.9686 ± 32.0264	**51.7226 ± 18.8433**	**93.5874 ± 36.4954**	**0.002 #**	**−41.8648 ± 26.9869**
12,13-EpOME	30.9825 ± 26.7698	20.7024 ± 11.0768	0.46	9.7907 ± 27.7448	**46.9109 ± 15.0402**	**79.7224 ± 23.6841**	**<0.001**	**−32.8115 ± 15.3972**
9,10-DiHOME	4.1455 ± 1.7899	3.2993 ± 1.7108	0.085	0.7458 ± 0.9866	5.5991 ± 2.7877	5.6467 ± 2.4652	0.954 #	−0.0476 ± 2.7904
12,13-DiHOME	4.9809 ± 1.9437	3.8968 ± 1.6289	0.161 #	0.9906 ± 1.6559	6.9594 ± 3.7873	6.9340 ± 3.3774	0.875 #	0.0254 ± 2.8341
**LOX metabolites**								
5-HETE	10.1945 ± 2.9568	8.9008 ± 2.5521	0.273 #	1.3211 ± 1.9205	11.6499 ± 3.0694	11.2376 ± 3.5321	0.53 #	0.4123 ± 3.1451
8-HETE	3.1627 ± 0.8646	2.9701 ± 0.9371	0.46 #	0.1954 ± 0.4666	3.5068 ± 1.3221	2.9656 ± 1.1078	0.185	0.5413 ± 1.3245
9-HETE	6.2378 ± 2.4916	5.0812 ± 2.0444	0.235 #	1.1362 ± 1.8189	6.6115 ± 2.5626	5.7522 ± 2.1829	0.209 #	0.8593 ± 2.1644
11-HETE	**4.3689 ± 1.3795**	**3.4094 ± 1.1668**	**0.027**	0.9124 ± 0.8519	4.5454 ± 1.4621	4.0433 ± 1.4575	0.071 #	0.5021 ± 1.5746
12-HETE	5.1343 ± 1.7200	5.8545 ± 2.6463	0.452 #	−0.7233 ± 1.8003	**6.0063 ± 2.3467**	**4.7972 ± 2.3597**	**0.023 #**	**1.2091 ± 1.8582**
15-HETE	6.1984 ± 1.9171	4.9051 ± 1.5388	0.085	1.2806 ± 1.1112	6.1869 ± 1.8919	5.7417 ± 3.2394	0.06 #	0.4452 ± 2.9529
4-HDHA	2.3149 ± 0.6472	2.1630 ± 0.5977	0.565 #	0.1653 ± 0.5413	2.4431 ± 0.9794	2.5609 ± 1.2943	0.875 #	−0.1178 ± 0.8863
7-HDHA	2.1060 ± 0.6072	1.7319 ± 0.5209	0.11	0.3892 ± 0.4553	2.1783 ± 0.8143	2.0061 ± 0.6765	0.259	0.1722 ± 0.5011
8-HDHA	1.2263 ± 0.4487	1.0900 ± 0.3821	0.44	0.1357 ± 0.1780	1.2990 ± 0.5943	1.1141 ± 0.4935	0.182 #	0.1849 ± 0.5362
10-HDHA	0.7721 ± 0.2512	0.6877 ± 0.2059	0.387	0.0793 ± 0.0990	0.7940 ± 0.3187	0.7020 ± 0.2701	0.196	0.0920 ± 0.2312
11-HDHA	1.0995 ± 0.3769	1.0493 ± 0.3586	0.902	0.0545 ± 0.2055	1.0862 ± 0.4291	1.0103 ± 0.3687	0.136 #	0.0758 ± 0.4079
13-HDHA	0.9168 ± 0.2861	0.8168 ± 0.2303	0.364	0.0963 ± 0.1441	0.9169 ± 0.3765	0.8015 ± 0.3238	0.259 #	0.1154 ± 0.3361
14-HDHA	1.1312 ± 0.4384	1.0678 ± 0.4203	0.727	0.0541 ± 0.2125	1.1712 ± 0.5441	0.9981 ± 0.5062	0.084 #	0.1732 ± 0.3398
16-HDHA	1.1430 ± 0.3468	0.9559 ± 0.3037	0.182	0.1782 ± 0.1994	1.1459 ± 0.4227	1.0653 ± 0.4675	0.117 #	0.0806 ± 0.4293
17-HDHA	1.4069 ± 0.4032	1.1633 ± 0.3601	0.141	0.2414 ± 0.2051	**1.4601 ± 0.5780**	**1.2773 ± 0.5100**	**0.041 #**	**0.1828 ± 0.5168**
20-HDHA	2.8698 ± 0.7707	2.4726 ± 0.7079	0.212 #	0.3802 ± 0.4872	2.9700 ± 0.9600	2.7939 ± 1.1603	0.084 #	0.1761 ± 0.9553
5-HEPE	1.4608 ± 0.5576	1.3111 ± 0.4406	0.481	0.1804 ± 0.3543	1.5968 ± 0.9534	1.4799 ± 0.6337	0.875 #	0.1170 ± 0.8054
8-HEPE	0.2118 ± 0.0669	0.1959 ± 0.0421	0.712	0.0193 ± 0.0428	0.2061 ± 0.1012	0.1817 ± 0.0752	0.329 #	0.0244 ± 0.0829
9-HEPE	0.4018 ± 0.1460	0.3661 ± 0.1209	0.528	0.0400 ± 0.1177	0.3664 ± 0.2280	0.3335 ± 0.1318	0.875 #	0.0329 ± 0.1871
11-HEPE	0.2752 ± 0.0907	0.2520 ± 0.0448	0.538	0.0217 ± 0.0697	0.2531 ± 0.1150	0.2364 ± 0.0928	0.388 #	0.0167 ± 0.1077
12-HEPE	0.5039 ± 0.1762	0.5213 ± 0.1704	0.812	−0.0162 ± 0.1369	0.4619 ± 0.2576	0.4038 ± 0.1731	0.638 #	0.0580 ± 0.2070
15-HEPE	0.3150 ± 0.1086	0.3142 ± 0.0634	0.983	0.0029 ± 0.0850	0.2890 ± 0.1524	0.2596 ± 0.1144	0.433 #	0.0294 ± 0.1342
18-HEPE	0.8580 ± 0.2946	0.8418 ± 0.2239	0.883 #	0.0156 ± 0.2704	0.7714 ± 0.3717	0.7502 ± 0.4476	0.53 #	0.0212 ± 0.3909
9-HODE	22.0631 ± 7.3185	17.1239 ± 6.0628	0.091 #	4.7896 ± 3.9253	36.2855 ± 21.5709	34.0942 ± 21.6186	0.071 #	2.1913 ± 11.7721
13-HODE	**17.7783 ± 6.0021**	**13.5081 ± 4.2496**	**0.012**	4.0845 ± 3.1626	25.0606 ± 10.8578	25.1650 ± 13.2212	0.48 #	−0.1044 ± 8.7534
**CYP ω/(ω−1)** **metabolites**								
16-HETE	0.2346 ± 0.0507	0.2175 ± 0.0583	0.462 #	0.0186 ± 0.0590	0.2194 ± 0.0562	0.2184 ± 0.0828	0.875 #	0.0010 ± 0.0660
17-HETE	0.0565 ± 0.0112	0.0560 ± 0.0145	0.922 #	0.0010 ± 0.0109	0.0602 ± 0.0182	0.0517 ± 0.0147	0.071 #	0.0085 ± 0.0161
18-HETE	0.1620 ± 0.0463	0.1499 ± 0.0282	0.538 #	0.0154 ± 0.0455	0.1651 ± 0.0450	0.1493 ± 0.0421	0.272 #	0.0158 ± 0.0416
19-HETE	0.1441 ± 0.0810	0.1354 ± 0.0411	0.854	0.0133 ± 0.0689	0.1599 ± 0.0533	0.1662 ± 0.0571	0.693	−0.0064 ± 0.0546
20-HETE	0.4738 ± 0.2011	0.3992 ± 0.2211	0.408 #	0.0787 ± 0.1668	0.4803 ± 0.1680	0.4787 ± 0.2014	0.958	0.0016 ± 0.1021
22-HDHA	0.0994 ± 0.0832	0.0763 ± 0.0603	0.325	0.0216 ± 0.0282	0.1067 ± 0.0758	0.1115 ± 0.0824	0.695 #	−0.0048 ± 0.0562
20-HEPE	0.1460 ± 0.0808	0.1343 ± 0.0692	0.713 #	0.0119 ± 0.0403	0.1550 ± 0.0963	0.1485 ± 0.0768	0.608	0.0065 ± 0.0426

Notes: A, arterial blood; V, venous blood; A-V difference, arteriovenous difference. Bold indicates significant difference.

**Table 3 metabolites-12-00034-t003:** Effects of hemodialysis on total plasma oxylipins and their ratios in the CKD patients before (pre-HD) and at cessation (post-HD) of hemodialysis (*n* = 12 each).

Amount ng/ml	Pre-HD A (Mean ± SD)	Pre-HD V (Mean ± SD)	*p*-Value, *t*-Test (# Paired Wilcoxon test)	Pre-HD A-V Difference (Mean ± SD)	Post-HD A (Mean ± SD)	Post-HD V (Mean ± SD)	*p*-Value, *t*-test (# Paired Wilcoxon Test)	Pre-HD A-V Difference (Mean ± SD)
5,6-EET + 5,6-DHET	26.3853 ± 22.7488	17.8740 ± 15.3357	0.325 #	8.2771 ± 22.0947	**38.4409 ± 16.7998**	**63.1169 ± 21.0481**	**0.003 #**	**−24.6760 ± 17.5393**
8,9-EET + 8,9-DHET	12.2661 ± 10.8593	8.8769 ± 6.7288	0.758 #	3.2897 ± 11.3985	**14.9171 ± 6.2221**	**23.6504 ± 7.8856**	**0.003 #**	**−8.7333 ± 7.0409**
11,12-EET + 11,12-DHET	10.3905 ± 10.0509	7.4546 ± 6.8226	0.902 #	2.7353 ± 10.2061	**14.7525 ± 6.7118**	**26.1620 ± 9.1754**	**0.002 #**	**−11.4095 ± 7.5222**
14,15-EET + 14,15-DHET	12.9774 ± 12.5015	10.0376 ± 9.5061	0.667 #	2.6766 ± 12.4408	**19.5791 ± 9.2852**	**36.1083 ± 13.6146**	**0.002 #**	**−16.5292 ± 10.9098**
7,8-EDP + 7,8-DiHDPA	4.9725 ± 4.3744	3.7040 ± 3.1733	0.854 #	1.2292 ± 4.8036	**6.4969 ± 2.8052**	**10.8874 ± 3.5963**	**0.002 #**	**−4.3904 ± 2.2808**
10,11-EDP + 10,11-DiHDPA	4.6062 ± 4.3839	3.4580 ± 3.0889	0.806 #	1.0598 ± 4.5591	**6.7795 ± 3.0931**	**12.0623 ± 3.9527**	**<0.001**	**−5.2828 ± 2.5347**
13,14-EDP + 13,14-DiHDPA	3.4752 ± 3.4913	2.7467 ± 2.8205	0.854 #	0.6542 ± 3.6762	**4.9215 ± 2.2192**	**7.9218 ± 2.6423**	**<0.001**	**−3.0002 ± 1.9422**
16,17-EDP + 16,17-DiHDPA	3.0618 ± 2.8699	2.2873 ± 2.0344	0.712 #	0.7529 ± 2.8441	**3.9502 ± 1.7125**	**5.6394 ± 2.0354**	**0.009**	**−1.6892 ± 1.8365**
19,20-EDP + 19,20-DiHDPA	6.3937 ± 5.2944	4.7947 ± 3.1021	1 #	1.5121 ± 5.6027	**9.6295 ± 4.0652**	**17.2258 ± 6.4814**	**<0.001**	**−7.5963 ± 3.8630**
5,6-EEQ + 5,6-DiHETE	1.5350 ± 0.9296	1.3729 ± 0.7133	0.622 #	0.3217 ± 0.8716	2.0440 ± 1.5009	1.7043 ± 0.8338	0.388 #	0.3397 ± 1.2857
8,9-EEQ + 8,9-DiHETE	1.5386 ± 1.2263	1.0880 ± 0.6966	0.285	0.4408 ± 1.3934	**2.7148 ± 1.8146**	**4.4861 ± 1.6000**	**0.008 #**	**−1.7714 ± 1.6800**
11,12-EEQ + 11,12-DiHETE	1.0017 ± 0.8179	0.7552 ± 0.4846	0.758 #	0.2422 ± 0.9263	**1.8673 ± 1.2279**	**3.5323 ± 1.3814**	**0.003 #**	**−1.6651 ± 1.1940**
14,15-EEQ + 14,15-DiHETE	0.9963 ± 0.7071	0.7508 ± 0.4514	0.328	0.2305 ± 0.8200	**1.8082 ± 1.2226**	**3.3214 ± 1.3339**	**0.004 #**	**−1.5133 ± 1.3083**
17,18-EEQ + 17,18-DiHETE	1.8998 ± 1.3591	1.6056 ± 0.8496	0.536	0.2613 ± 1.5047	**3.2405 ± 2.1560**	**6.5059 ± 2.3420**	**0.003 #**	**−3.2654 ± 2.0580**
9,10-EpOME + 9,10-DiHOME	37.1683 ± 31.3783	25.8981 ± 13.6496	0.712 #	10.7144 ± 32.1530	**57.3217 ± 18.9513**	**99.2342 ± 38.1881**	**0.002 #**	**−41.9125 ± 28.5066**
12,13-EpOME + 12,13-DiHOME	35.9634 ± 27.1475	24.5992 ± 12.1650	0.424 #	10.7813 ± 27.9218	**53.8703 ± 15.3169**	**86.6564 ± 25.6785**	**<0.001**	**−32.7861 ± 16.4280**
5,6-DHET/5,6-EET	0.0749 ± 0.0518	0.0864 ± 0.0565	0.617		**0.0431 ± 0.0261**	**0.0222 ± 0.0120**	**0.002 #**	
8,9-DHET/8,9-EET	0.2737 ± 0.2578	0.2858 ± 0.2519	0.854 #		**0.1253 ± 0.0630**	**0.0599 ± 0.0480**	**0.002 #**	
11,12-DHET/11,12-EET	0.0827 ± 0.0649	0.0874 ± 0.0650	0.926 #		**0.0426 ± 0.0205**	**0.0198 ± 0.0099**	**0.002 #**	
14,15-DHET/14,15-EET	0.0654 ± 0.0409	0.0550 ± 0.0284	0.485		**0.0326 ± 0.0210**	**0.0127 ± 0.0047**	**0.002 #**	
7,8-DiHDPA/7,8-EDP	0.1650 ± 0.1420	0.5745 ± 1.4719	0.622 #		**0.0892 ± 0.0353**	**0.0458 ± 0.0285**	**0.002 #**	
10,11-DiHDPA/10,11-EDP	0.0538 ± 0.0350	0.0514 ± 0.0285	0.859		**0.0256 ± 0.0118**	**0.0110 ± 0.0041**	**0.002 #**	
13,14-DiHDPA/13,14-EDP	0.0506 ± 0.0302	0.0484 ± 0.0269	0.855		**0.0237 ± 0.0101**	**0.0134 ± 0.0039**	**0.01 #**	
16,17-DiHDPA/16,17-EDP	0.0552 ± 0.0392	0.0601 ± 0.0343	0.755		**0.0335 ± 0.0139**	**0.0194 ± 0.0071**	**0.008**	
19,20-DiHDPA/19,20-EDP	0.2730 ± 0.2232	0.2905 ± 0.1983	0.806 #		**0.1212 ± 0.0480**	**0.0580 ± 0.0264**	**0.002 #**	
5,6-DiHETE/5,6-EEQ	483.3002 ± 348.2197	599.0058 ± 431.5977	0.49		**274.7175 ± 163.7055**	**121.8920 ± 63.0904**	**0.001**	
8,9-DiHETE/8,9-EEQ	0.1037 ± 0.0654	0.1332 ± 0.0872	0.372		**0.0502 ± 0.0261**	**0.0229 ± 0.0107**	**0.002 #**	
11,12-DiHETE/11,12-EEQ	0.1107 ± 0.1548	0.0634 ± 0.0260	0.806 #		**0.0596 ± 0.1111**	**0.0112 ± 0.0032**	**0.002 #**	
14,15-DiHETE/14,15-EEQ	0.1012 ± 0.0933	0.0817 ± 0.0318	0.854 #		**0.0436 ± 0.0466**	**0.0153 ± 0.0044**	**0.003 #**	
17,18-DiHETE/17,18-EEQ	0.2320 ± 0.1551	0.2159 ± 0.1591	0.667 #		**0.1399 ± 0.1461**	**0.0397 ± 0.0133**	**0.002 #**	
9,10-DiHOME/9,10-EpOME	0.1799 ± 0.0860	0.1731 ± 0.1090	0.58 #		**0.1260 ± 0.0918**	**0.0617 ± 0.0222**	**0.01 #**	
12,13-DiHOME/12,13-EpOME	0.2255 ± 0.0981	0.2087 ± 0.0951	0.356 #		**0.1627 ± 0.1101**	**0.0877 ± 0.0327**	**0.008 #**	
Ratio(5,6-DHET+8,9-DHET+11,12-DHET+14,15-DHET) / (5,6-EET+8,9-EET +11,12 EET +14,15-EET)	0.1104 ± 0.0878	0.1188 ± 0.0946	1 #		**0.0541 ± 0.0262**	**0.0253 ± 0.0143**	**0.002 #**	
Ratio(7,8-DiHDPA+10,11-DiHDPA +13,14-DiHDPA+16,17-DiHDPA+19,20-DiHDPA) / (7,8-EDP+10,11-EDP+13,14-EDP+16,17-EDP+19,20-EDP)	0.1369 ± 0.1095	0.1478 ± 0.1106	0.902 #		**0.0666 ± 0.0240**	**0.0336 ± 0.0143**	**0.002 #**	
Ratio(5,6-DiHETE+8,9-DiHETE+11,12-DiHETE+14,15-DiHETE+17,18-DiHETE) / (5,6-EEQ+ 8,9-EEQ+11,12-EEQ+14,15-EEQ+17,18-EEQ)	0.6541 ± 0.4424	0.6335 ± 0.4128	0.909		**0.3403 ± 0.2022**	**0.1315 ± 0.0660**	**0.002 #**	
Ratio (9,10-DiHOME+12,13-DiHOME) / (9,10-EpOME+12,13-EpOME)	0.2017 ± 0.0845	0.1904 ± 0.1007	0.58 #		**0.1445 ± 0.1017**	**0.0741 ± 0.0258**	**0.008 #**	

Notes: A, arterial blood; V, venous blood; A-V difference, arteriovenous difference. Bold indicates significant difference.

**Table 4 metabolites-12-00034-t004:** Effects of hemodialysis on free oxylipins in the CKD patients at cessation (post-HD) of hemodialysis (*n* = 12 each).

	Post-HD A (Mean ± SD)	Post-HD V (Mean ± SD)	*p*-Value, *t*-Test (# Paired Wilcoxon Test)	Post-HD A-V Difference (Mean ± SD)
**CYP epoxy-metabolites**				
5,6-EET	0.2022 ± 0.0733	0.2317 ± 0.0812	0.303	−0.0295 ± 0.0945
8,9-EET	0.2672 ± 0.1586	0.2517 ± 0.1676	0.814 #	0.0155 ± 0.1947
11,12-EET	0.1754 ± 0.0675	0.2403 ± 0.1094	0.051	−0.0649 ± 0.1029
14,15-EET	0.2573 ± 0.1199	0.3791 ± 0.2067	0.049	−0.1219 ± 0.1913
5,6-DHET	0.0324 ± 0.0085	0.0321 ± 0.0104	0.814 #	0.0002 ± 0.0047
8,9-DHET	0.0910 ± 0.0281	0.0847 ± 0.0241	0.388 #	0.0063 ± 0.0207
11,12-DHET	0.1486 ± 0.0443	0.1454 ± 0.0379	0.631	0.0032 ± 0.0226
14,15-DHET	0.2038 ± 0.0569	0.2062 ± 0.0477	0.854	−0.0024 ± 0.0444
7,8-EDP	0.1852 ± 0.1197	0.2412 ± 0.1309	0.076	−0.0560 ± 0.0993
10,11-EDP	0.1358 ± 0.0948	0.1805 ± 0.1004	0.079	−0.0447 ± 0.0800
13,14-EDP	0.1273 ± 0.0808	0.1452 ± 0.0769	0.439	−0.0179 ± 0.0774
16,17-EDP	0.1922 ± 0.0843	0.2284 ± 0.0893	0.124	−0.0362 ± 0.0753
19,20-EDP	0.2899 ± 0.1669	0.3587 ± 0.1754	0.096	−0.0688 ± 0.1311
7,8-DiHDPA	0.0234 ± 0.0210	0.0330 ± 0.0115	0.388 #	−0.0097 ± 0.0267
10,11-DiHDPA	0.0581 ± 0.0148	0.0554 ± 0.0125	0.182 #	0.0027 ± 0.0067
13,14-DiHDPA	0.0696 ± 0.0222	0.0661 ± 0.0178	0.272 #	0.0035 ± 0.0090
16,17-DiHDPA	0.0865 ± 0.0293	0.0809 ± 0.0206	0.695 #	0.0056 ± 0.0148
19,20-DiHDPA	0.7042 ± 0.3142	0.6061 ± 0.2823	0.06 #	0.0981 ± 0.1753
8,9-EEQ	0.0663 ± 0.0419	0.0700 ± 0.0284	0.791	−0.0037 ± 0.0472
11,12-EEQ	0.0435 ± 0.0354	0.0490 ± 0.0252	0.388 #	−0.0055 ± 0.0418
14,15-EEQ	0.0607 ± 0.0349	0.0655 ± 0.0298	0.651	−0.0049 ± 0.0363
17,18-EEQ	0.0830 ± 0.0590	0.0936 ± 0.0518	0.602	−0.0106 ± 0.0684
5,6-DiHETE	0.0980 ± 0.0508	0.0799 ± 0.0403	0.01	0.0181 ± 0.0322
8,9-DiHETE	0.0233 ± 0.0063	0.0202 ± 0.0041	0.174	0.0031 ± 0.0074
11,12-DiHETE	0.0160 ± 0.0026	0.0155 ± 0.0023	0.461	0.0004 ± 0.0020
14,15-DiHETE	0.0181 ± 0.0080	0.0167 ± 0.0073	0.525	0.0015 ± 0.0078
17,18-DiHETE	0.1661 ± 0.0704	0.1476 ± 0.0603	0.337	0.0185 ± 0.0640
9,10-EpOME	4.6969 ± 3.1153	6.2952 ± 5.0363	0.182 #	−1.5984 ± 3.6422
12,13-EpOME	6.8441 ± 3.8513	8.4182 ± 6.5171	0.239 #	−1.5740 ± 4.1754
9,10-DiHOME	1.7695 ± 1.1345	1.7400 ± 1.1665	0.754 #	0.0295 ± 0.2442
12,13-DiHOME	3.3260 ± 2.0952	3.4845 ± 2.2350	0.347 #	−0.1585 ± 0.4498
**LOX metabolites**				
5-HETE	0.1100 ± 0.0465	0.1147 ± 0.0494	0.365	−0.0046 ± 0.0169
8-HETE	0.0672 ± 0.0281	0.0667 ± 0.0298	0.943	0.0006 ± 0.0273
9-HETE	0.1016 ± 0.0158	0.1058 ± 0.0218	0.31	−0.0042 ± 0.0136
11-HETE	0.0739 ± 0.0265	0.0800 ± 0.0305	0.262	−0.0061 ± 0.0180
12-HETE	0.5660 ± 0.3630	0.4319 ± 0.2449	0.094	0.1341 ± 0.2531
15-HETE	0.1789 ± 0.0583	0.1841 ± 0.0689	0.536	−0.0052 ± 0.0282
4-HDHA	0.0821 ± 0.0209	0.0844 ± 0.0239	0.695 #	−0.0023 ± 0.0109
7-HDHA	0.0852 ± 0.0189	0.0875 ± 0.0186	0.594	−0.0024 ± 0.0148
8-HDHA	0.0779 ± 0.0221	0.0763 ± 0.0241	0.584	0.0016 ± 0.0098
10-HDHA	0.0622 ± 0.0148	0.0636 ± 0.0173	0.658	−0.0014 ± 0.0105
11-HDHA	0.0722 ± 0.0107	0.0766 ± 0.0117	0.081	−0.0044 ± 0.0079
13-HDHA	0.0653 ± 0.0131	0.0671 ± 0.0143	0.272 #	−0.0018 ± 0.0065
14-HDHA	0.1165 ± 0.0638	0.1098 ± 0.0627	0.308 #	0.0067 ± 0.0218
16-HDHA	0.0589 ± 0.0142	0.0610 ± 0.0122	0.389	−0.0022 ± 0.0083
17-HDHA	0.1191 ± 0.0301	0.1196 ± 0.0325	0.924	−0.0005 ± 0.0174
20-HDHA	0.2074 ± 0.0539	0.2079 ± 0.0489	0.347 #	−0.0004 ± 0.0370
5-HEPE	0.1032 ± 0.0315	0.0962 ± 0.0336	0.171	0.0070 ± 0.0166
8-HEPE	0.0352 ± 0.0045	0.0327 ± 0.0051	0.128	0.0025 ± 0.0053
9-HEPE	0.0756 ± 0.0137	0.0703 ± 0.0095	0.155	0.0053 ± 0.0121
11-HEPE	0.0847 ± 0.0025	0.0856 ± 0.0035	0.875 #	−0.0008 ± 0.0045
12-HEPE	0.1595 ± 0.0220	0.1540 ± 0.0228	0.136 #	0.0055 ± 0.0158
15-HEPE	0.1078 ± 0.0123	0.1036 ± 0.0105	0.084 #	0.0042 ± 0.0085
18-HEPE	0.1779 ± 0.0413	0.1596 ± 0.0278	0.07 #	0.0183 ± 0.0316
9-HODE	8.5190 ± 13.8940	8.3464 ± 12.8721	0.347 #	0.1726 ± 1.8733
13-HODE	4.3069 ± 3.0614	4.4488 ± 2.9851	0.53 #	−0.1419 ± 0.6334
**CYP ω/(ω−1) metabolites**				
16-HETE	0.0626 ± 0.0120	0.0631 ± 0.0114	0.772	−0.0006 ± 0.0069
17-HETE	0.0242 ± 0.0028	0.0240 ± 0.0029	0.713	0.0002 ± 0.0018
18-HETE	0.0456 ± 0.0082	0.0451 ± 0.0072	0.684	0.0005 ± 0.0044
19-HETE	0.0387 ± 0.0291	0.0392 ± 0.0269	0.638 #	−0.0005 ± 0.0216
20-HETE	0.1457 ± 0.1239	0.0709 ± 0.0885	0.084 #	0.0747 ± 0.1464
22-HDHA	0.0843 ± 0.0662	0.0754 ± 0.0632	0.177	0.0089 ± 0.0213
20-HEPE	0.0900 ± 0.0271	0.0856 ± 0.0284	0.638 #	0.0045 ± 0.0171

Notes: A, arterial blood; V, venous blood; A-V difference, arteriovenous difference.

## Data Availability

No new data were created or analyzed in this study. Data sharing is not applicable to this article.

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
