# Peer review of "Hemodialysis and Plasma Oxylipin Biotransformation in Peripheral Tissue"

_metabolites, 2022, doi:10.3390/metabo12010034_

Round 1

Reviewer 1 Report

The manuscript „Hemodialysis and plasma oxylipin biotransformation in peripheral tissue“ by Tong Liu, Inci Dogan, Michael Rothe, Julius V. Kunz, Felix Knauf, Maik Gollasch, Friedrich C. Luft and Benjamin Gollasch presents new data showing that hemodialysis produces specific changes, in particular negative arteriovenous differences, in CYP-epoxy metabolites not observed before hemodialysis.  

This referee has the following comments:

Major points:

  1. General: Please check for typos (e.g. missing articles, super- and subscript).
  2. Introduction: While a clear hypothesis is presented it remains unclear why hemodialysis should affect the arteriovenous difference of the metabolites studied. Is a mechanism known that may explain this effect? Please clarify.
  3. Fig. 2: It is not quite clear where the peripheral compartments are shown. What is meant by the words on the left side shown together with the circled arrow (e.g. uremia, dialyzer etc.)?
  4. Table1: When data are given as x.y, all other data should be given in the same format. Alternatively, the decimal place can be omitted.
  5. Table S1 & S2 & S3, Table 2 & 3 & 4: In the legend it should read ±SD
  6. Tables: You may consider to provide the data with significant differences in another colour (and not only in bold black) for easier orientation. Please check numbers – in some places the ± sign is given not before the SD-value
  7. Table 2: Please discuss whether the small number of significant differences pre-HD among a large list of non-significant data may be just false positive results. Do you consider it necessary to correct for the multiple comparisons made, for example by adjusting the p-value threshold (similar to the Bonferoni correction)? Especially consider the effect size in these cases (AV difference) and explain whether the difference found may have any functional relevance. In addition, when effect size was small and no difference was detected, may this be due to an n being too small?
  8. Results, pre-HD: Please consider whether it is useful to list all metabolites without a difference in the text again, while they are all shown in table 2.
  9. Results, p.8: There are two Dunn's multiple comparison test reported. It is not clear what is the difference between them.
  10. Results, post-HD: You state that “However, the effects were stronger in venous blood”. How did you define “stronger”, which numbers have been compared and what statistics used to come to this conclusion? The same question relates to your statement “The data show that all four classes of CYP epoxy-metabolites (i.e. 182 EET, EpOME, EEQ and EDP plus their respective diols) were stronger accumulated in 183 venous blood compared to arterial blood”. Please clarify.
  11. Methods, Determination of eicosanoid profiles: The methods should be described briefly, reference only to published papers is not sufficient.

Author Response

Reviewer 1:

Comments to the Author

The manuscript „Hemodialysis and plasma oxylipin biotransformation in peripheral tissue“ by Tong Liu, Inci Dogan, Michael Rothe, Julius V. Kunz, Felix Knauf, Maik Gollasch, Friedrich C. Luft and Benjamin Gollasch presents new data showing that hemodialysis produces specific changes, in particular negative arteriovenous differences, in CYP-epoxy metabolites not observed before hemodialysis. 

Our response: Thank you for your positive comments. We revised our manuscript and fully addressed all issues indicated in the review report.

  1. General: Please check for typos (e.g. missing articles, super- and subscript).

Our response: Thank you. We have thoroughly reviewed the manuscript for typos.

  1. Introduction: While a clear hypothesis is presented it remains unclear why hemodialysis should affect the arteriovenous difference of the metabolites studied. Is a mechanism known that may explain this effect? Please clarify.

Our response: We were particularly interested to understand how the plasma oxylipins are modified by extracorporeal renal replacement therapy, which is known to cause oxidative stress, chronic inflammation and red blood cell-endothelial interactions. We were aware that Fick first capitalized on the arteriovenous oxygen difference to determine cardiac output and thereby to determine oxygen delivery and extraction. His insights permitted measuring the amount of oxygen taken up from the blood by the individual tissues. Usually, the arterial oxygen concentration is measured in blood from the femoral, brachial, or radial artery, and the oxygen content of mixed venous blood is measured from blood withdrawn from the right heart (i.e. pulmonary artery). The mixed venous oxy-gen content represents the weighted average of oxygen content in venous blood from all organ systems. However, venous blood can also be withdrawn from specific organs, e.g. legs, arms or the mesentery. In this case, measurement of oxygen consumption by the specific organ system requires determination of the oxygen content of venous blood draining that organ. We applied this approach to better understand biotransformation of oxylipins in vivo, particularly whether or not the peripheral tissues, especially the muscles in the upper limbs, either produce, store or degrade part of the epoxy-metabolites that pass through them in response to dialysis treatment. Our data indicate that CYP epoxy-metabolites are influenced by renal-replacement therapies, and are consistent with the notion that blood perfusing peripheral tissue, especially the muscle, acts as a stimulus for release and accumulation of CYP epoxy-metabolites in the upper limb during dialysis treatment. Based on our data, we suggest that CYP epoxy-metabolites may be a contributing factor to the blood flow response in the peripheral circulation during hemodialysis. Future studies can clarify whether the identified metabolites exhibit beneficial or detrimental cardiovascular effects, possibly in metabolite-interacting networks. These interrelationships have been discussed in great detail in the Discussion section. Please note that this approach has been used by other authors to study ammonia levels, glucose metabolism and NO formation. We stated this clearly the Introduction section and provided references.

  1. Fig. 2: It is not quite clear where the peripheral compartments are shown. What is meant by the words on the left side shown together with the circled arrow (e.g. uremia, dialyzer etc.)?

Our response:  According to your comment, we now better explain the meanings of the circled arrows in the text and changed the legend of Figure 2.

  1. Table1: When data are given as x.y, all other data should be given in the same format. Alternatively, the decimal place can be omitted.

Our response: Thank you. Done.

  1. Table S1 & S2 & S3, Table 2 & 3 & 4: In the legend it should read ±SD

Our response: Tables have been changed according to your suggestions.

  1. Tables: You may consider to provide the data with significant differences in another colour (and not only in bold black) for easier orientation. Please check numbers – in some places the ± sign is given not before the SD-value

Our response: We agree. Colour has been changed and ± sign is now thoroughly given in the text.

  1. Table 2: Please discuss whether the small number of significant differences pre-HD among a large list of non-significant data may be just false positive results. Do you consider it necessary to correct for the multiple comparisons made, for example by adjusting the p-value threshold (similar to the Bonferoni correction)? Especially consider the effect size in these cases (AV difference) and explain whether the difference found may have any functional relevance. In addition, when effect size was small and no difference was detected, may this be due to an n being too small?

Our response: Please note that we chose a significance level of 0.05, which means we assume a 5 % chance of error that our results could be false positive. As to the functional relevance, please see our comprehensive discussion about known biological effects of the mediators. We admit that the research field is new, and the technological requirements to measure such mediators are novel, so there is still little data on the pathophysiological significance of these mediators. Our paper is a first step to better understand the role of oxylipins in CKD and their biotransformation in vivo. Please note that we used Dunn's multiple comparison test to compare the diol/epoxide ratios of the epoxy-metabolites (Table 3), because here we compared four groups. In all other comparisons we did not have to adjust the p-values to more than two groups.

  1. Results, pre-HD: Please consider whether it is useful to list all metabolites without a differe nce in the text again, while they are all shown in table 2.

Our response: We appreciate  that the tables are comprehensive. We believe that it will be easier for our readers to follow our manuscript if we list key metabolites, even if they do not change. Please note that we did not note “all” metabolites that have not been changed in the main text.

  1. Results, p.8: There are two Dunn's multiple comparison test reported. It is not clear what is the difference between them.

Our response: Please note we have used Dunn's multiple comparison test only once. However, we have emphasized the meaning twice in our manuscript, so that the reader can better follow the significance of our findings.

  1. Results, post-HD: You state that “However, the effects were stronger in venous blood”. How did you define “stronger”, which numbers have been compared and what statistics used to come to this conclusion? The same question relates to your statement “The data show that all four classes of CYP epoxy-metabolites (i.e. 182 EET, EpOME, EEQ and EDP plus their respective diols) were stronger accumulated in 183 venous blood compared to arterial blood”. Please clarify.

Our response:  Thank you. Text has been changed.

  1. Methods, Determination of eicosanoid profiles: The methods should be described briefly, reference only to published papers is not sufficient.

Our response: We agree. Additions have been made to the methods section based on your comments. Please see the revised version of the manuscript.

Reviewer 2 Report

This is an interesting study on the impact of HD on lipid metabolism with special focus on oxylipins. The data are valuable.

Only a few aspects require clarification.

Patient characteristics

The exact inclusion/exclusion criteria for the patients should be given. The group is small and contains no diabetics, although this is the largest group of patients on HD. This should be clarified.

The mean age of the group suggests that only the elderly were included. Was it on purpose?

The patients had 3 HD sessions per week. During which HD session the blood was taken – the one after a longer break or that in the middle of the week?

All patients had hyperlipidemia. The information on statin treatment and potential impact of such therapy on oxylipins should be given.

Results

The group is small, thus high variability of obtained  values is observed, with SD values comparable or even exceeding mean values. It should be commented on.

The number of data is overwhelming and tables become sometimes difficult to follow. Wouldn’t it be better to concentrate on statistically significant results and put the others into the supplementary materials?

Do you think we should consider oxylipins uremic toxins? Please comment.

Author Response

Reviewer 2:

Comments to the Author

This is an interesting study on the impact of HD on lipid metabolism with special focus on oxylipins. The data are valuable. Only a few aspects require clarification.

Our response: Thank you for your positive comments. We appreciate the positive criticism and made appropriate changes.

Patient characteristics:

  1. The exact inclusion/exclusion criteria for the patients should be given. The group is small and contains no diabetics, although this is the largest group of patients on HD. This should be clarified. The mean age of the group suggests that only the elderly were included. Was it on purpose?

Our response: Thank you for your comment. The inclusion and exclusion criteria have now been made clearer. Our study includes a patient population that is characterized by an irreversible impairment of kidney function requiring chronic hemodialysis treatment. The design of our study does not differentiate between patient groups undergoing long-term dialysis therapy with regard to the specific underlying renal disease or age. It was a coincidence that our study did not include patients with diabetic nephropathy leading to end stage renal disease. The age of our patients is not surprisingly high, as in Germany the mean age of HD patients is 78 years (data of the QIN registry of the KfH, G. von Gersdorff and colleagues, University of Cologne, 2015).

  1. The patients had 3 HD sessions per week. During which HD session the blood was taken – the one after a longer break or that in the middle of the week?

Our response:  We took random blood samples during the week with the aim of observing the impact of a single dialysis treatment but not to evaluate the impact of the interdialytic interval.

  1. All patients had hyperlipidemia. The information on statin treatment and potential impact of such therapy on oxylipins should be given.

Our response: Thank you. Four of our patients were under statin treatment; from the pathophysiological context, we assume no significant effect of statin treatment on the arteriovenous difference of plasma oxylipins. Instead, we were focused on the impact of a single dialysis treatment on plasma levels of oxylipins in our study.

Results

  1. The group is small, thus high variability of obtained values is observed, with SD values comparable or even exceeding mean values. It should be commented on.

Our response:  Please note that we chose a significance level of 0.05, which means we assume a 5 % chance of error that our results could be false positive or false negative. We used paired t-tests for all data that strictly obeyed normal distribution, otherwise paired non-parametric tests were used. Descriptive statistics were obtained, and variables were checked for skewness and kurtosis to ensure that they met the normal distribution assumptions.

  1. The number of data is overwhelming and tables become sometimes difficult to follow. Wouldn’t it be better to concentrate on statistically significant results and put the others into the supplementary materials?

Our response:  We respectfully disagree because in this case the reader might lose the overview for the systematic approach. By now color-coding the statistically significant results, readers can follow the paper more easily (See also our comments to Reviewer 1).

  1. Do you think we should consider oxylipins uremic toxins? Please comment.

Our response: We have not compared uremic patients with non-uremic patients in our study.

Instead, we have studied the impact of a single HD treatment on plasma levels of oxylipins in CKD patients. To clarify whether oxylipids can be considered as uremic toxins, a different study design would have been necessary, including different patient groups of CKD, patients with diabetic nephropathy, different age and gender, all adjusted to control groups etc.
